# Federated Learning for Human-in-the-Loop Many-to-Many Voice Conversion

*Ryunosuke Hirai, Yuki Saito, and Hiroshi Saruwatari*

The University of Tokyo, Japan.

ryu1999827@gmail.com, yuuki_saito@ipc.i.u-tokyo.ac.jp

## Abstract

We propose a method for training a many-to-many voice conversion (VC) model that can additionally learn users' voices while protecting the privacy of their data. Conventional many-to-many VC methods train a VC model using a publicly available or proprietary multi-speaker corpus. However, they do not always achieve high-quality VC for input speech from various users. Our method is based on federated learning, a framework of distributed machine learning where a developer and users cooperatively train a machine learning model while protecting the privacy of user-owned data. We present a proof-of-concept method on the basis of StarGANv2-VC (i.e., Fed-StarGANv2-VC) and demonstrate that our method can achieve speaker similarity comparable to conventional non-federated StarGANv2-VC.

**Index Terms**: many-to-many voice conversion, federated learning, human-in-the-loop, distributed machine learning, StarGANv2-VC

## 1. Introduction

Voice conversion (VC) [1] is a technology that converts one speaker's voice characteristics into another speaker's while keeping the phonetic content of input speech unchanged. VC can enrich speech communication between humans through speech representation beyond the physical constraints of human vocal organs, such as speaking aid for voice disorders [2]. From this perspective, VC technology requires a framework that can handle diverse speakers as both the source and target of VC and achieve high naturalness of the converted speech and speaker similarity to the target speaker.

Non-parallel many-to-many VC, the main focus of this study, is a machine learning framework for training a single VC model that can convert any arbitrary speakers included in the training data (i.e., seen speakers). The main advantage of this framework is its scalability with respect to the number of seen speakers because it does not necessarily require parallel data with identical phonetic content for all speakers. A StarGAN-VC series [3, 4] and variational autoencoder (VAE)-based methods [5, 6] are examples of non-parallel many-to-many VC using deep generative models such as generative adversarial network (GAN) [7] and VAE [8]. In particular, StarGANv2-VC [9] can accurately convert not only speaker identity but also speaking style of input speech and thus potentially handle a diverse range of users' voices.

In a typical application scenario of many-to-many VC, a developer first trains a VC model using a publicly available or proprietary multi-speaker corpus and then provides users with a VC application using the model, as shown in Fig. 1(a). However, such *one-way* approach cannot guarantee the conversion accuracy for input speech data from various users without any

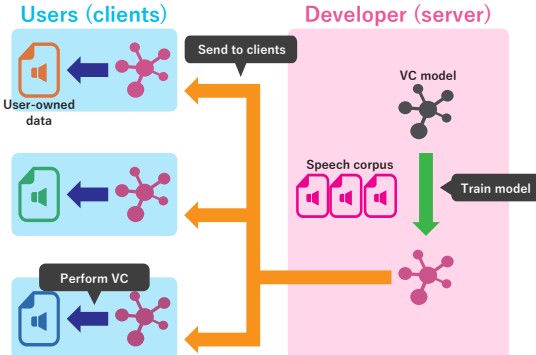

**(a) Conventional one-way approach for many-to-many VC**

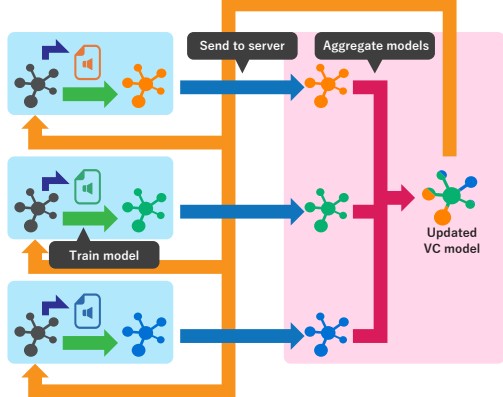

**(b) Proposed two-way approach for many-to-many VC**

Figure 1: *Many-to-many VC with (a) conventional and (b) proposed approaches.*

techniques for few-/zero-shot speaker adaptation [10, 11]. If sequential updates to a developer-provided VC model with user-owned data while keeping the data privacy were possible, one could realize a *two-way* approach for many-to-many VC where users can join the VC model training and offer the developer a way to improve the diversity of speakers who can be converted by the VC model.

To this end, we propose a federated learning method for a non-parallel many-to-many VC model that can additionally learn users' voices while protecting their data privacy. In the proposed method, a developer and users act as a central server and distributed clients, respectively, and iterate 1) sending a VC model on the server to randomly selected clients, 2) training the model using each user's data, and 3) aggregating model parameters sent by the clients to the server. We present a proof-of-concept method on the basis of StarGANv2-VC (i.e., *Fed-*

*StarGANv2-VC*) and evaluate its performance on a VC task using monolingual multi-speaker corpus. The evaluation results demonstrate that our method can achieve speaker similarity comparable to conventional non-federated StarGANv2-VC.

## 2. Baseline Method: StarGANv2-VC

StarGANv2-VC [9] improves upon StarGAN-VC [3] and enables high-quality non-parallel many-to-many VC of diverse speakers. This section describes the components of the StarGANv2-VC model.

### 2.1. Style encoder

A style encoder $S(\boldsymbol{X}_{\mathrm{ref}}, c)$ takes a reference speech sample $\boldsymbol{X}_{\mathrm{ref}}$ spoken by the $c$th speaker as the input and generates the speaker's style vector $\boldsymbol{s}$. This continuous vector representation improves the diversity of speaking styles reproducible by a StarGANv2-VC generator better than the discrete speaker ID $c$ used in StarGAN-VC for conditioning the generator.

### 2.2. Mapping network

In addition to the aforementioned style encoder, a mapping network $M(\boldsymbol{z}, c)$ is simultaneously trained to generate the $c$th speaker's style vector from a Gaussian noise $\boldsymbol{z} \sim \mathcal{N}(\boldsymbol{0}, \boldsymbol{I})$. This allows for random sampling of style vectors without using the target speaker's reference speech and models the stochastic variation of speaking styles.

### 2.3. StarGANv2-VC generator

A generator $G(\cdot)$, which converts one speaker's speech (or its mel-spectrogram) $\boldsymbol{X}$ into another speaker's, consists of the following three sub-modules:

1. **Encoder** extracts a latent vector $\boldsymbol{h}_{\mathrm{x}}$ from $\boldsymbol{X}$.
2. **F0 network** extracts a prosodic feature vector $\boldsymbol{h}_{\mathrm{F0}}$ from $\boldsymbol{X}$.
3. **Decoder** takes $\boldsymbol{h}_{\mathrm{x}}$, $\boldsymbol{h}_{\mathrm{F0}}$, and the target speaker's style vector $\boldsymbol{s}$ as inputs, and reconstructs the mel-spectrogram $\hat{\boldsymbol{X}}$ of the target speaker.

Li et al. [9] use a pre-trained Joint Detection and Classification (JDC) network [12] as the F0 network to ensure that the F0 contour pattern of the source speaker remains invariant during the VC process.

### 2.4. StarGANv2-VC discriminators

The StarGANv2-VC training involves two classification models: a real/fake discriminator $D(\cdot)$ and a source classifier $C(\cdot)$. The $D(\cdot)$ and $C(\cdot)$ learn the speaker-specific difference between natural and converted speech samples and an input speaker's identity from the given mel-spectrogram, respectively. These two networks have a stack of shared layers that extracts a discriminative feature vector from an input mel-spectrogram and take the vector as input to predict the natural/converted label and speaker ID.

### 2.5. Multi-task adversarial training

The objective function of the aforementioned networks can be regarded as an adversarial training between the generator and discriminators. The discriminator $D(\cdot)$ and classifier $C(\cdot)$ are trained to discriminate between natural/converted speech samples and to classify the source speaker's identity from input mel-spectrogram, respectively. In contrast, the generator $G(\cdot)$ is updated to deceive these networks by generating mel-spectrogram that can be discriminated as the VC target speaker's natural speech while considering some regularization loss functions, such as cycle-consistency [13], style restoration/diversification,

---

**Algorithm 1** Server-side algorithm in FedAvg
---
1: $\mathcal{C} = \{1, 2, \ldots, K\} :=$ the set of client indices
2: $M :=$ the number of randomly selected clients
3: $\boldsymbol{w}^k :=$ parameters of a model owned by the $k$th client
4: $n^k :=$ the number of data owned by the $k$th client
5: $R :=$ the number of rounds for FL
6: **for** $r = 1, 2, \ldots, R$ **do**
7: $\quad \mathcal{C}_r \in \mathcal{C} :=$ the set of indices for randomly selected clients at the $r$th round ($|\mathcal{C}_r| = M$)
8: $\quad N_r = \sum_{k \in \mathcal{C}_r} n^k :=$ the total number of data
9: $\quad$ **for all** $k \in \mathcal{C}_r$ **do**
10: $\quad\quad$ Send the global model to the $k$th client: $\boldsymbol{w}_r^k \leftarrow \boldsymbol{w}_r$
11: $\quad\quad$ Update on the $k$th client $\boldsymbol{w}_r^k \leftarrow \mathrm{Update}(k, \boldsymbol{w}_r^k)$
12: $\quad$ **end for**
13: $\quad$ Aggregate the local models: $\boldsymbol{w}_r \leftarrow \sum_{k \in \mathcal{C}_r} \frac{n^k}{N_r} \boldsymbol{w}_r^k$
14: **end for**

---

**Algorithm 2** Client-side algorithm in FedAvg Update($k, \boldsymbol{w}$)
---
1: $\mathcal{D}^k :=$ the set of data owned by the $k$th client
2: $\mathcal{B}^k :=$ the set of mini-batches taken from $\mathcal{D}^k$
3: $E :=$ the number of local epoch
4: $\eta :=$ learning rate
5: $h(b; \boldsymbol{w}) :=$ a loss function calculated by the model parameters $\boldsymbol{w}$ and mini-batch $b$
6: **for** $i = 1, 2, \cdots, E$ **do**
7: $\quad$ **for** $b \in \mathcal{B}^k$ **do**
8: $\quad\quad \boldsymbol{w} \leftarrow \boldsymbol{w} - \eta \boldsymbol{\nabla}_{\boldsymbol{w}} h(b; \boldsymbol{w})$
9: $\quad$ **end for**
10: **end for**
11: Send the model parameters $\boldsymbol{w}$ to the server

---

phonetic-content consistency, and prosody feature (e.g., F0 and norm) consistency.

## 3. Proposed Method: Fed-StarGANv2-VC

In this study, we aim to realize human-in-the-loop many-to-many VC while protecting the privacy of user-owned data, and propose Fed-StarGANv2-VC by applying the federated learning framework to StarGANv2-VC.

### 3.1. Federated learning

Federated learning (FL) [14] is a type of distributed machine learning framework where a developer and users work as server-client that cooperatively trains a machine learning model while protecting the privacy of user-owned data. We first briefly explain the concept of FL.

**FL round:** The server and clients iterate the following steps called "round" during the FL process:

1. The server sends the parameters of a global model to randomly selected multiple clients.
2. The selected clients use their own data to train the received model for a certain period (local epoch).
3. The clients send the trained model parameters, not their own data, to the server.
4. The server aggregates the models sent by the clients and updates its own global model.

Since the server and clients send and receive only the model parameters during FL, the trained global model can learn the users' data without violating their privacy.

**Federated averaging (FedAvg):** FedAvg [15] is a widely used model aggregation method in FL. In FedAvg, the aggregated model parameters are calculated by computing the

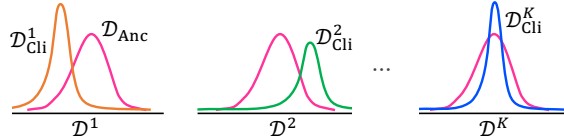

Figure 2: *Illustration of non-iid nature of our FL setting.*

weighted average of the model parameters sent by clients, whose weight coefficients are given by the number of data each client has. Algorithms 1 and 2 describe the FedAvg algorithms on server and client sides, respectively.

### 3.2. Fed-StarGANv2-VC algorithm

In conventional non-parallel many-to-many VC methods, a developer first trains a VC model using a publicly available or proprietary multi-speaker corpus and then provides users with it for realizing a VC application. Users can fine-tune the provided model with their own data to improve the performance of VC using the data. As we explained in Section 1, this one-way approach cannot offer a way to improve the VC model owned by the developer with the users.

**Dataset definition:** To realize human-in-the-loop training for a many-to-many VC model, we define the following two datasets:

- **Anchor** $\mathcal{D}_{\mathrm{Anc}}$ is accessible by the developer and all clients.
- **Client** $\mathcal{D}_{\mathrm{Cli}}^{k}$ is accessible by only the $k$th client.

The $\mathcal{D}_{\mathrm{Anc}}$ and $\mathcal{D}_{\mathrm{Cli}}^{k}$ represent a publicly available multi-speaker corpus for learning a speaker space shared among the clients and one user's private data, respectively. In our FL, the $k$th client can use $\mathcal{D}^k = \mathcal{D}_{\mathrm{Anc}} \cup \mathcal{D}_{\mathrm{Cli}}^{k}$ for calculating the loss function $h(b; \boldsymbol{w})$ and its gradient with respect to the model parameters.

**Server/Client update:** The basic algorithm for training our Fed-StarGANv2-VC follows FedAvg described in Algorithms 1 and 2. Specifically, a VC developer (i.e., server) first distributes an initial StarGANv2-VC model and Anchor dataset $\mathcal{D}_{\mathrm{Anc}}$ to randomly selected $M$ users (i.e., clients). Then, the selected clients update the distributed StarGANv2-VC model parameters using their own devices and the dataset $\mathcal{D}^k$ consisting of the Anchor dataset and the clients' private data. After this local update, the server receives the $M$ model parameters from the selected clients and aggregates them to update the global VC model parameter. This iterative update enables the VC model to increase the diversity of speakers whose voices can be converted by the model without sharing the clients' private voice data.

**Countermeasure for non-iid nature:** In our FL setting, the independent and identically distributed (iid) nature of the training data is no longer guaranteed. This is because the distribution of the data owned by each client is expected to be different from each other, as shown in Fig. 2, and potentially causes the trained model to overfitting for a specific client in FL. To deal with this issue, we introduce FedProx [16] that prevent the client model parameter $\boldsymbol{w}_r^k$ from deviating too much from the server model parameter $\boldsymbol{w}_r$ and considers a regularization term expressed as $\mu\|\boldsymbol{w}_r^k - \boldsymbol{w}_r\|^2$ during the training, where $\mu \in [0, 1]$ is a hyperparameter that controls the effect of this regularization.

### 3.3. Discussion

Our FL-based many-to-many VC method has several hyperparameters that can affect the performance of a trained VC model. In Section 4, we investigate some hyperparameter settings for the number of clients for aggregating model parameters, the number of FL round, and the use of FedProx regularization. We leave the investigation of the effects caused by dataset sizes (i.e., the numbers of Anchor/Client speakers and utterances per speaker) future work.

In FL, one has to consider attacks [17] by malicious users trying to make the training unstable or disclose the identity of user(s) from the learned global model. For example, poisoning attacks contaminate training data (e.g., injecting noisy samples) [18] or change the parameters of a global model directly [19]. Although the experiments in this paper assumes a setting where a malicious attacker does not participate in FL, further study is necessary to introduce countermeasures to deal with the attacker and to prevent the leakage of users' biometric information.

## 4. Experimental Evaluation

### 4.1. Experimental setup

In this experiment, we compared the conventional non-federated StarGANv2-VC method [9] with our Fed-StarGANv2-VC using a monolingual multi-speaker corpus and evaluated their VC performances on the basis of objective and subjective evaluations. Although the server and client in the actual FL setting are assumed to be different computers or edge devices, we conducted our FL experiment on a virtual server and client in the same computer for simplicity.

We used an open-sourced implementation of StarGANv2-VC[1] published by the first author of [9]. The neural network architecture and speech parameter extraction settings were the same as this implementation, and the neural vocoder for synthesizing a speech waveform from a mel-spectrogram was a pre-trained Parallel WaveGAN [20].

We used the "parallel100" subset of the JVS corpus [21] sampled at 24,000 Hz and randomly selected 40 speakers (20 males and 20 females) from the corpus. After concatenating all of the speech samples of each of these speakers, we first removed the silence intervals of 100 ms or longer from the concatenated speech and then divided the silence-removed speech into 5-second segments, which were used as a single data unit. The numbers of training, validation, and test data were 3,284, 411, and 411, respectively. To reproduce the situation where each client has speech data of different speakers, the 40 speakers were divided into 10 "Anchor" speakers (Anc) and 30 "Client" speakers (Cli). The number of each Anchor speaker's speech data and the Client speaker's speech data were generally balanced.

For the FL setting, we set the local epoch to 10. The batch-size was 10. The optimizer was AdamW [22]. The FedProx hyperparameter $\mu$ was set to 1.

Each of the following evaluations was conducted separately for the four VC settings (i.e., {Anc, Cli}-to-{Anc, Cli}) when the source/target speaker was an Anchor speaker or a Client speaker. The difficulty of VC for each setting is different, and in particular, VC between Client speakers (Cli-to-Cli) is considered to be the most difficult because it cannot be learned directly by FL. To simulate the situation where the users' voice data were unavailable for the server, we generated the target speaker's style vector using the mapping network and fed it to the VC model during inference.

---

[1] https://github.com/yl4579/StarGANv2-VC

Table 1: *x-vector cosine similarity results for our Fed-StarGANv2-VC (mean±std)*

| # Cli | Model # round | FedProx? | Anc→Anc | Anc→Cli | Cli→Anc | Cli→Cli |
|---|---|---|---|---|---|---|
| 1 | 200 | No | 0.48 ± 0.19 | 0.23 ± 0.39 | 0.48 ± 0.18 | 0.23 ± 0.39 |
| 1 | 400 | No | 0.51 ± 0.17 | 0.37 ± 0.40 | 0.51 ± 0.17 | 0.38 ± 0.40 |
| 3 | 200 | No | 0.53 ± 0.19 | 0.26 ± 0.41 | 0.54 ± 0.20 | 0.27 ± 0.41 |
| 3 | 400 | No | 0.53 ± 0.16 | 0.39 ± 0.37 | 0.53 ± 0.17 | 0.41 ± 0.36 |
| 1 | 200 | Yes | 0.53 ± 0.20 | 0.31 ± 0.38 | 0.53 ± 0.21 | 0.32 ± 0.38 |
| 1 | 400 | Yes | 0.52 ± 0.17 | 0.41 ± 0.36 | 0.52 ± 0.18 | 0.43 ± 0.35 |
| 3 | 200 | Yes | **0.56 ± 0.19** | 0.34 ± 0.36 | **0.56 ± 0.19** | 0.35 ± 0.36 |
| 3 | 400 | Yes | 0.53 ± 0.18 | 0.42 ± 0.35 | 0.53 ± 0.18 | 0.43 ± 0.34 |
| 3 | 600 | Yes | 0.55 ± 0.18 | 0.44 ± 0.35 | 0.55 ± 0.19 | 0.46 ± 0.34 |
| 3 | 800 | Yes | 0.55 ± 0.19 | **0.46 ± 0.35** | 0.55 ± 0.19 | **0.48 ± 0.34** |

Table 2: *x-vector cosine similarity results for conventional non-federated (i.e., baseline) StarGANv2-VC (mean±std)*

| # Epoch | Anc→Anc | Anc→Cli | Cli→Anc | Cli→Cli |
|---|---|---|---|---|
| 200 | 0.49 ± 0.21 | 0.50 ± 0.34 | 0.49 ± 0.22 | 0.51 ± 0.34 |
| 300 | 0.54 ± 0.17 | 0.52 ± 0.33 | 0.55 ± 0.17 | 0.53 ± 0.32 |
| 400 | 0.51 ± 0.20 | 0.52 ± 0.35 | 0.53 ± 0.20 | 0.53 ± 0.34 |
| 500 | 0.52 ± 0.18 | 0.50 ± 0.35 | 0.52 ± 0.17 | 0.50 ± 0.34 |
| 600 | **0.55 ± 0.19** | 0.55 ± 0.32 | 0.55 ± 0.18 | **0.56 ± 0.32** |
| 700 | 0.54 ± 0.18 | **0.55 ± 0.33** | **0.56 ± 0.18** | 0.55 ± 0.33 |
| 800 | 0.53 ± 0.19 | 0.53 ± 0.33 | 0.54 ± 0.18 | 0.53 ± 0.33 |

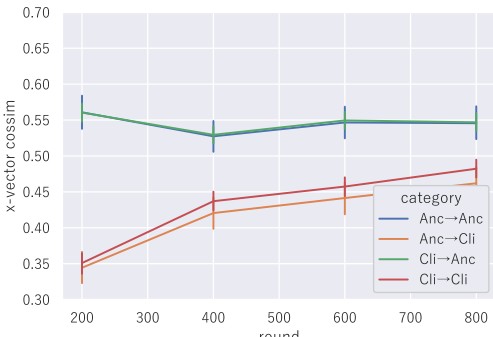

Figure 3: *x-vector cosine similarity (x-vector cossim) curves against the increase of the number of rounds in FL with model aggregation from three clients and FedProx regularization.*

## 4.2. Objective evaluation

We used the cosine similarity between x-vectors [23] extracted from the target speaker's speech samples and ones converted by the many-to-many VC model. For the x-vector extraction, we used a open-sourced model[2] pretrained by using the JTube-Speech [24] corpus.

### 4.2.1. Investigation of FL parameter settings

We investigated the performance of our Fed-StarGANv2-VC in terms of 1) the number of clients used for the aggregation process (1 or 3), 2) the number of rounds (from 200 to 800), and 3) the FedProx regularization term in FL. The evaluation results of the x-vector cosine similarity for Fed-StarGANv2-VC are shown in Table 1. From the results, we observed the following trends.

- The evaluation results consistently improved by increasing the number of clients for aggregating model parameters.
- The evaluation results generally tended to improve as the

[2]https://github.com/sarulab-speech/xvector_jtubespeech

number of rounds increased, which is also shown in Fig. 3, especially in the VC cases that used the Client speakers (Cli) as the target (red and orange lines).
- The effect of FedProx was small when the Anchor speaker was the VC target, and a significant improvement was observed in the opposite case.

These results suggest that 1) the model parameter aggregation from multiple clients is effective to increase the diversity of speakers handled by the VC model, 2) by iterating sufficient FL rounds, the model can even learn VC into Client speakers, and 3) a countermeasure for the non-iid nature of FL is also important for the non-parallel many-to-many VC task. In the subsequent evaluations, we used Fed-StarGANv2-VC trained using the parameters { # Cli = 3, # round = 800, with FedProx regularization } because it achieved the highest x-vector cosine similarity in VC into the Client speakers.

### 4.2.2. Investigation of non-federated StarGANv2-VC

To compare StarGANv2-VC models trained without FL, we calculated the x-vector cosine similarity for different numbers of epochs (from 200 to 800). The evaluation results are shown in Table 2. Note that the same four conversion speaker pairs as Fed-StarGANv2-VC were used, but the VC model did not distinguish between {Anc, Cli} during the training. From the table, we confirm that the x-vector cosine similarity tends to improve with an increase in the number of epochs for all the VC cases. In addition, although there was no significant difference in the objective metrics for 600 and 700 epochs, a slight degradation was observed at 800 epochs. Therefore, for the subsequent evaluations, we used the StarGANv2-VC model trained for 700 epochs for the pairwise comparison to our Fed-StarGANv2-VC model.

## 4.3. Subjective evaluations

We conducted two subjective evaluations: a preference XAB test on the similarity to the target speaker and a preference AB test on the naturalness of converted speech. In these evalua-

Table 3: *Preference XAB scores on speaker similarity to the target speaker*

| VC setting | Conventional | Proposed |
|---|---|---|
| Cli(F) → Anc(F) | 0.448 | **0.552** |
| Cli(F) → Anc(M) | 0.484 | 0.516 |
| Cli(M) → Anc(F) | 0.432 | **0.568** |
| Cli(M) → Anc(M) | 0.490 | 0.510 |
| Cli(F) → Cli(F) | 0.520 | 0.480 |
| Cli(F) → Cli(M) | 0.472 | 0.528 |
| Cli(M) → Cli(F) | 0.510 | 0.490 |
| Cli(M) → Cli(M) | 0.480 | 0.520 |

Table 4: *Preference AB scores on speech naturalness*

| VC setting | Conventional | Proposed |
|---|---|---|
| Cli(F) → Anc(F) | 0.526 | 0.474 |
| Cli(F) → Anc(M) | **0.574** | 0.426 |
| Cli(M) → Anc(F) | **0.584** | 0.416 |
| Cli(M) → Anc(M) | **0.604** | 0.396 |
| Cli(F) → Cli(F) | 0.366 | **0.634** |
| Cli(F) → Cli(M) | 0.368 | **0.632** |
| Cli(M) → Cli(F) | 0.326 | **0.674** |
| Cli(M) → Cli(M) | 0.408 | **0.592** |

tions, we randomly selected three test data from each of the 30 Client speakers and converted them into the other speakers for evaluating whether the trained VC models could handle users' voices. The total number of evaluations was 2 (XAB or AB) × 2 (Cli-to-{Anc, Cli}) × 4 ({male, female}-to-{male, female}) = 16, and we recruited 50 listeners for each VC case through crowdsourcing on Lancers[3]. Therefore, the total number of listeners was $50 \times 16 = 800$. In the XAB test, we first presented the target speaker's natural speech as reference to the listeners and asked them to answer which speech sample sounded more similar to the target speaker. Each listener evaluated 10 pairs of converted speech samples.

The subjective evaluation results for speaker similarity and naturalness are shown in Tables 3 and 4, respectively. The gender of input and output speakers is represented as (M) for male and (F) for female. Bold indicates that the method is significantly better than the other on the basis of the Student's t-test ($p < 0.05$). From Table 3, the proposed method achieved speaker similarity that is equal to or significantly higher than that of the conventional method for all VC settings. This result suggests that the proposed method has the potential to improve the diversity of speakers through FL. One reason for this improvement might be that the conventional method tried to learn VC between 40 speakers at once with a single model, while the proposed method made the learning easier since each client dataset contains only 1 Client speaker and 10 Anchor speakers.

In contrast, Table 4 showed a significant difference in the naturalness of the converted speech depending on the VC settings. Specifically, compared to the conventional method, the naturalness of speech samples converted by the proposed method was significantly lower in Cli-to-Anc VC and significantly higher in Cli-to-Cli VC. One reason might be the data imbalance of the Client and Anchor speakers. In other words, there were more Client speakers (30) than Anchor speakers (10) and the minority was not sufficiently learned by the FL for many-to-many VC.

---

³https://www.lancers.jp/

## 5. Conclusion

In this study, we proposed federated learning (FL) for non-parallel many-to-many voice conversion (VC) that can learn users' voices while protecting the privacy of their recordings. We presented a proof-of-concept method using a StarGANv2-VC model and evaluated its performance through the objective and subjective evaluations. The results showed that even for VC between Clients (i.e., users), which could not be directly learned through FL, our proposed method achieved the speaker similarity comparable to the conventional non-federated method. In future work, we plan to investigate in detail the effects of the data imbalance between Anchor/Client speakers and the numbers of these speakers. In addition, we will evaluate the proposed method on different devices, rather than virtually.

**Acknowledgements:** This work was supported by Research Grant A and Research Grant S of the Tateishi Science and Technology Foundation.

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
