# OpenReview forum: "Federated Learning for Human-in-the-Loop Many-to-Many Voice Conversion"
_Interspeech.org/2023/Workshop/SSW — SSW12_

### Official Review · Reviewer_KgQB · 2023-06-04
**Federated Learning for Human-in-the-Loop Many-to-Many Voice Conversion**

**Rating:** 8
**Confidence:** 4

**Review:**

-Key Strength of the paper

A VC architecture is proposed based on a federated learning scheme, in which updates of local models are made on distributed clients then aggregated in a single model to the server without requiring to collect the data of the speakers.

Two experiments are conducted, both objectively by monitoring the x-vector cosine similarity, and subjectively by conducting a similarity and a naturalness tests.

-Main Weakness of the paper

There is an inconsistency between the space used to describe the original starGANv2-VC and for the description of the proposed federated learning scheme.

For the objective experiment, why don't you provide the the cosine similarity mesures for the baseline starGANv2-VC?

It is not clear to me what assumptions the authors make on their proposed FL-VC? The main point is for me the fact that it does not require to centralise the data to update the VC. What are the reasons why the similarity or the similarity could improve?

In particular, I wonder whether the experimental results are not biased by the relatively small number of speakers used. The observations that are made in Section 4.2 may not be valid anymore with an increased number of anchor speakers (for instance 100 or 1,000). Also, the maximum number of clients used in the experiments is relatively small (3)

-Novelty/Originality, taking into account the relevance of the work for the SSW audience

Federated VC is certainly original and relevant, especially regard the issue of voice and data privacy

-Technical Correctness, is the work technically and/or scientifically solid? Are sufficient details provided to allow any experiments to be reproduced or equivalent experiments run?

The authors provide statistical significancy tests, but there is no explanation about how it is actually processed.

---

> ### Author Response · Authors · 2023-06-28
> **Author responses to Reviewer KgQB**
>
> First of all, thank you for reading our manuscript and giving valuable comments.
>
> -----
>
> Comment 1: "There is an inconsistency between the space used to describe the original starGANv2-VC and for the description of the proposed federated learning scheme."
>
> Response 1: In accordance with this comment, we shortened the explanation about the original StarGANv2-VC written in Section 2.
>
> -----
>
> Comment 2: "For the objective experiment, why don't you provide the the cosine similarity mesures for the baseline starGANv2-VC?"
>
> Response 2: We apologize for confusing you, but we provided the cosine similarity for the baseline StarGANv2-VC in Table 2.
>
> -----
>
> Comment 3: "It is not clear to me what assumptions the authors make on their proposed FL-VC? The main point is for me the fact that it does not require to centralise the data to update the VC. What are the reasons why the similarity or the similarity could improve?"
>
> Response 3: Our main assumption was that the decentralized learning of a many-to-many VC model can sequentially improve the speaker diversity covered by the model through the interaction between the developer (server) and users (clients). As shown in Table 3, our federated VC even achieved better speaker similarity than the non-federated VC, although it suffered from the degradation of speech naturalness (Table 4). As described in the second and third paragraphs of Section 4.3, we attributed the reasons to 1) the simplified VC learning due to data decentralization and 2) data imbalance of the numbers of Anchor and Client speakers, but further experiments with larger speech corpora are needed.
>
> -----
>
> Comment 4: "In particular, I wonder whether the experimental results are not biased by the relatively small number of speakers used. The observations that are made in Section 4.2 may not be valid anymore with an increased number of anchor speakers (for instance 100 or 1,000). Also, the maximum number of clients used in the experiments is relatively small (3)"
>
> Response 4: Thank you for your suggestion. We plan to conduct larger scale experiment and thorough investigation of the hyperparameter settings in FL.
>
> -----
>
> Comment 5: "The authors provide statistical significancy tests, but there is no explanation about how it is actually processed."
>
> Response 5: We conducted Student's t-test. We added this explanation in the second paragraph of the camera-ready paper.

---

### Official Review · Reviewer_jRXe · 2023-06-13
**This paper presents one of the first studies on the application of federated learning for the purpose of improving voice conversion quality for new speakers in a way that doesn't involve breaching privacy or fully retraining models. Results show that there is an improvement on the achievable speaker similarity thanks to the distributed learning, and we can expect follow up work based on this preliminary study to polish the idea further.**

**Rating:** 7
**Confidence:** 4

**Review:**

Strengths:

Fairly novel study on the application of federated learning for the purpose of distributed customer-privacy preserving VC technologies that learns with time. Even if the VC system isn't novel that doesn't diminish the impact and value of this research.

Improvement areas:

Section 2 is over-formulated in my oppinion. While it is good to be comprehensive, half a page of formulas that can be consulted in a different paper doesn't add to the content of the paper and it is hard to justify given the limited space and other gaps in the paper.

With the statistics presented in Tables 1 and 2, is it possible to claim statistical significance? (as a minor comment, I found it weird personally to mention mean+-std and not mean+-confidence margin). This is alleviated by having the preference tests in 4.3 though.

Comments/questions

How can we guarantee that we are protecting the privacy of the customer at inference time in this situation? In a 'cyberattack' situation in which the client can be forced to generate any speaker embedding at conversion time, would it be possible to make sure that customer voices aren't recoverable in thsi framework, and if so, how? Otherwise, it would be better to call out that what we are maintaining is the privacy of the speech recordings and not of the biometric recoverability.

There are grammatical issues here and there, but there was one that was quite distractign as it is not clear what it means and it breaks the claims in the conclusion: 'our proposed method achieved the speaker similarity the conventional method' what was achieved, better similarity? the same?

---

> ### Author Response · Authors · 2023-06-28
> **Author responses to Reviewer jRXe**
>
> First of all, thank you for reading our manuscript and giving valuable comments.
>
> -----
>
> Comment 1: "Section 2 is over-formulated in my oppinion. While it is good to be comprehensive, half a page of formulas that can be consulted in a different paper doesn't add to the content of the paper and it is hard to justify given the limited space and other gaps in the paper."
>
> Response 1: In accordance with this comment, we shortened the explanation about the original StarGANv2-VC written in Section 2.
>
> -----
>
> Comment 2: "With the statistics presented in Tables 1 and 2, is it possible to claim statistical significance? (as a minor comment, I found it weird personally to mention mean+-std and not mean+-confidence margin). This is alleviated by having the preference tests in 4.3 though."
>
> Response 2: We did not conduct statistical significance tests for the objective evaluation results because the purpose here was to investigate better models for the conventional and proposed models trained with various hyperparameter settings.
>
> -----
>
> Comment 3: "How can we guarantee that we are protecting the privacy of the customer at inference time in this situation? In a 'cyberattack' situation in which the client can be forced to generate any speaker embedding at conversion time, would it be possible to make sure that customer voices aren't recoverable in thsi framework, and if so, how? Otherwise, it would be better to call out that what we are maintaining is the privacy of the speech recordings and not of the biometric recoverability."
>
> Response 3: Thank you for your very insightful comment. As you guessed, our main focus in this paper is to protect the privacy of the users' speech recordings rather than their biometric information. We revised the manuscript so that the readers can easily understand this focus. Since the topic of this paper is many-to-many VC, the proposed federated-learning-based VC model can reproduce the clients' voices through the update of global/local models, as shown in Tables 3 and 4. From this perspective, we added the discussion of the proposed VC framework including possible threat of the privacy leakage (i.e., attacks by malicious clients) in Section 3.3.
>
> -----
>
> Comment 4: "There are grammatical issues here and there, but there was one that was quite distractign as it is not clear what it means and it breaks the claims in the conclusion: 'our proposed method achieved the speaker similarity the conventional method' what was achieved, better similarity? the same?"
>
> Response 4: Thank you for pointing out the typos. We proofread the manuscript and corrected typos and grammatical errors.

---

### Official Review · Reviewer_CLjf · 2023-06-14
**Proof of concept, very limited contributions**

**Rating:** 5
**Confidence:** 3

**Review:**

The authors present a proof of concept kind of study about the application of a Federated Learning strategy for Voice Conversion.

The motivation is to evaluate a proof of concept of a VC system that will progressively learn from user's data by keeping the privacy of the data (two ways approach) at difference of a conventional approach where a developer provides user with a fixed VC system that would present particular challenges in terms of user's data privacy for model update.

The authors selected StarGANv2-VC as the VC system. The authors provided a detailed description of it including individual models and objective functions (this might not be necessary).

The authors then present the Federated Learning algorithmic strategy, no claims are presented/justified regarding the novelty of it.

It is mentioned that overfitting risks are possible since the the clients (users) data may not observe independent and identically distributed conditions and a regularization feature called FedProx is proposed to alleviate this potential issue.

Data from 40 speakers was used for the experimental evaluation. 10 were used as "anchor" speakers and the remaining 30 were used as "client" speakers. These number of speakers may not be sufficient for general performance observations. Additionally, since the FL setup based on a virtual server and client resources from the same computer the evaluation scope of the Federated Learning environment seems to be very restricted as well. Only a small number of clients was simulated (three, maximum).

The authors experimentally compared the proposed Federated Learning approach against a conventional setup using the same baseline VC system. The results in terms of objective cosine similarity and subjective similarity and naturalness show comparable performance.

The study overall suggests somehow the applicability of Federated Learning for non-parallel voice conversion but a larger study seems necessary to establish solid performance and computationally efficiency observations.

---

> ### Author Response · Authors · 2023-06-28
> **Author responses to Reviewer CLjf**
>
> First of all, thank you for reading our manuscript and giving valuable comments.
>
> -----
>
> Comment 1: "The authors selected StarGANv2-VC as the VC system. The authors provided a detailed description of it including individual models and objective functions (this might not be necessary)."
>
> Response 1: In accordance with this comment, we shortened the explanation about the original StarGANv2-VC written in Section 2.
>
> -----
>
> Comment 2: "Data from 40 speakers was used for the experimental evaluation. 10 were used as "anchor" speakers and the remaining 30 were used as "client" speakers. These number of speakers may not be sufficient for general performance observations. Additionally, since the FL setup based on a virtual server and client resources from the same computer the evaluation scope of the Federated Learning environment seems to be very restricted as well. Only a small number of clients was simulated (three, maximum)."
>
> Response 2: Thank you for your comment. Although the experimental results shown in this paper might be insufficient, we believe that our proposed federated-learning-based VC technology will open up a new research area of building better VC model based on human-machine interaction. We plan to conduct larger scale experiment and thorough investigation of the hyperparameter settings in FL.
>
> -----
>
> Comment 3: "The study overall suggests somehow the applicability of Federated Learning for non-parallel voice conversion but a larger study seems necessary to establish solid performance and computationally efficiency observations."
>
> Response 3: Thank you the comment. We will conduct additional experiments using different VC model architectures and VC tasks in the actual VC settings (i.e., physically distributed computers and devices) in our future study.

---

### Decision · Program_Chairs · 2023-06-14

**Decision:**

Accept

**Comment:**

SSW2003 received 45 papers. The acceptance rate is 82%. We are pleased to inform you that your paper has been accepted by the SSW2023 Program Committee. Please read the reviews carefully and submit your camera-ready paper by June 28th. Most reviewers performed a detailed review. Please answer to their questions and consider their comments. Note that camera-ready papers are credited with one extra page to allow authors to consider reviewers’ suggestions. So max 7 pages in total including figures & refs.
The deadline for submitting the revised version (with full non-anonymized authors and refs!) is 28th June.